# Microbial Diversity of a Disused Copper Mine Site (Parys Mountain, UK), Dominated by Intensive Eukaryotic Filamentous Growth

**DOI:** 10.3390/microorganisms10091694

**Published:** 2022-08-24

**Authors:** Marco A. Distaso, Rafael Bargiela, Bethan Johnson, Owen A. McIntosh, Gwion B. Williams, Davey L. Jones, Peter N. Golyshin, Olga V. Golyshina

**Affiliations:** 1Centre for Environmental Biotechnology, School of Natural Sciences, Bangor University, Deiniol Road, Bangor LL57 2UW, UK; 2Food Futures Institute, Murdoch University, 90 South Street, Perth, WA 6150, Australia

**Keywords:** Parys Mountain/Mynydd Parys, acid mine drainage environment, acidophilic bacteria, acidophilic archaea, Thermoplasmatales, *Ca*. Micrarchaeota, acidophilic eukaryotes

## Abstract

The Parys Mountain copper mine (Wales, UK) contains a wide range of discrete environmental microniches with various physicochemical conditions that shape microbial community composition. Our aim was to assess the microbial community in the sediments and overlying water column in an acidic mine drainage (AMD) site containing abundant filamentous biogenic growth via application of a combination of chemical analysis and taxonomic profiling using 16S rRNA gene amplicon sequencing. Our results were then compared to previously studied sites at Parys Mt. Overall, the sediment microbiome showed a dominance of bacteria over archaea, particularly those belonging to Proteobacteria (genera *Acidiphilium* and *Acidisphaera*), Acidobacteriota (subgroup 1), Chloroflexota (AD3 cluster), Nitrospirota (*Leptospirillum*) and the uncultured Planctomycetota/CPIa-3 termite group. Archaea were only present in the sediment in small quantities, being represented by the Terrestrial Miscellaneous Euryarchaeota Group (TMEG), Thermoplasmatales and *Ca*. Micrarchaeota (*Ca*. Micracaldota). Bacteria, mostly of the genera *Acidiphilium* and *Leptospirillum*, also dominated within the filamentous streamers while archaea were largely absent. This study found pH and dissolved solutes to be the most important parameters correlating with relative proportions of bacteria to archaea in an AMD environment and revealed the abundance patterns of native acidophilic prokaryotes inhabiting Parys Mt sites and their niche specificities.

## 1. Introduction

The large, abandoned copper mine Parys Mountain (Parys Mt; also known as Mynydd Parys) represents a site of high scientific, historic and cultural importance. Firstly, Parys Mt is an extreme environment with very low pH and high concentrations of heavy metals [1]. Secondly, the two rivers that drain through the mine (Afon Goch Dulas and Afon Goch Amlwch) deliver twice as much copper and zinc to the Irish Sea as all other water bodies of north Wales and northwest England combined (Mersey, Dee, Conwy, Ribble, Wyre and Duddon) [2]. Thirdly, the area possesses a unique geology, being the only example of Kuroko-type massive volcanic-associated sulfide deposits in the UK, as well as containing an abundance of the ore deposit anglesite (PbSO_4_), which was discovered at the site alongside a range of other Pb-, Zn- and Fe-containing minerals (e.g., pyrite, chalcopyrite, sphalerite, galena). Combined, these aspects make Parys Mt a place of great scientific interest for geochemical and extremophile studies [3,4,5]. Investigations have been undertaken to examine the microbial communities, the chemical and physical characteristics of the watercourses draining the site (e.g., Afon Goch Dulas) and the impact of acid mine drainage (AMD) on the surrounding wetlands [6,7,8,9]. Other microbial-based studies have focused on the biological communities present in an underground lake located at the site [10,11]. Further studies have addressed the growth of macroscopic biological structures (e.g., streamers) in the mine water and the microbial communities therein [7,11,12]. These works have established the taxonomic affiliations of prokaryotic groups inhabiting the mine site, highlighted the key role that iron-oxidizing and heterotrophic microorganisms play in biogeochemical cycling and determined the geochemical parameters that shape these unusual microbial communities. More recently, high-resolution metagenomic sequencing has been used to characterise the sediment and overlying aquatic communities at one of the most acidic sites at Parys Mt (denoted ”PM4”) [13]. Another study identified distinct microbial communities in the upper sediment layers (0–20 cm in depth), with a significant proportion of the community represented by archaea [14]. Furthermore, sequence signatures of organisms previously affiliated to “microbial dark matter” were identified [13,14,15]. In the context of extremophiles, a number of novel prokaryotic acidophilic microbial taxa were isolated, studied and described [8,15,16,17,18]. When all the inhabitants from sediments, planktonic and streamer-associated microbial communities were compared, the microbiomes were found to contain a diverse array of taxa, including Proteobacteriota, Actinobacteriota, Nitrospirae, Bacteroidetes, Acidobacteria, Firmicutes, Planctomycetota and Chloroflexi [7,10,11,13,14]. Among the archaea, representatives of the order Thermoplasmatales (phylum Euryarchaeota/*Ca*. Thermoplasmatota, class Thermoplasmata) were identified as a major group, together with less abundant taxa of organisms branching with the Terrestrial Miscellaneous Euryarchaeota Group (TMEG) and *Ca*. Micrarchaeota (*Ca*. Microcaldota) [13,14]. A further study used amplicon sequencing to assess eukaryotic diversity at nine chemically contrasting sites at Parys Mt [19]. This found that algal diversity was largely represented by the phylum Chlorophyta, within which the families Chlamydomonadaceae, Chlorellaceae, Coccomyxaceae and Koliellaceae were present. *Chlamydomonas acidophila* was found to dominate in most acidic and metal-rich ponds within Parys Mt [19]. Cultures of *Chlorella protothecoides* var. *acidicola* and *Euglena mutabilis* have since been isolated from Parys Mt [20].

In the context of extremophile microbiome discovery, the purpose of this work was to study the microbial communities at a biologically rich drainage site at Parys Mt (designated PM5). This site exhibits two key scientific features of interest; namely, (i) a very low pH in comparison to other sites at Parys Mt which we hypothesized would host hyperacidophilic microorganisms; and (ii) extensive filamentous biological growth that fills the mine drainage pool. The aims of this investigation were, therefore, to assess archaeal and bacterial taxonomic affiliations and relative abundances and to predict their metabolic potential. The prokaryotic community of PM5 was then compared to the community inhabiting the Parys Mt stream (PM4), which was recently analysed using the same methods, showing archaea to be prevalent. In this report, we discuss the microbial constituents of the mine adit site and their niche specificity along with the potential roles of microorganisms populating the abandoned Parys Mt copper mine.

## 2. Materials and Methods

### 2.1. Site Description and Sampling

Parys Mt is an abandoned copper mine (53.38747; −4.33968) on the island of Anglesey, northwest Wales, UK, a region with a humid temperate oceanic climate. Samples were collected in November 2020 from the adit (man-made horizontal passage leaving the mine filled with underground- and surface-water mine drainage). The sampling site was a shallow pool (depth 10−15 cm), which was the uppermost water body of a cascade of pools and ponds, hereinafter dubbed PM5. Samples contained sediments, saturated with water and a yellowish-green biofilm (filamentous acid streamer), growing in and filling the pool from the surface to the bottom (Figure 1). The adit was sampled in three locations in triplicate: (1) at the point of entrance of the water from the underground source into the pool (PM5.1); (2) in the central part of the pond (PM5.2); and (3) at the outflow edge of the pool, where the water system flows into the next basin of this cascade located at a level ca. 50 cm below the uppermost pond (PM5.3). Streamer growth was abundant at the inflow point and in the middle of the pool and visually reduced as it approached the outflow area, PM5.3. Additionally, 50 mL of samples of water-saturated sediments from the previously studied acidic stream PM4 [13,14] were collected on the same sampling day for comparative analysis. All samples were immediately delivered to the laboratory and kept at 4 °C until processing for molecular biological purposes (within 24 h) and chemical analysis. Temperature, pH and Eh (redox potential) were measured in the field using a SevenGo^®^ multimeter (Mettler-Toledo, Leicester, UK) in triplicate at each location.

### 2.2. Chemical Analysis

Sediment samples were oven-dried (105 °C, 24 h) to determine moisture content and ground to a fine powder before determining their total elemental composition using a S2 PicoFox TXRF Spectrometer (Bruker AXS Inc., Madison, WI, USA). The sediment pore water was recovered non-destructively using a 0.15 µm pore size Rhizon^®^ sampler (Rhizosphere Research Products, Wageningen, The Netherlands) before analysis of soluble elements using a Varian 720 ICP-OES. Pore water phosphate was determined using the colorimetric molybdate blue method from [21]. pH and electrical conductivity (EC) of the sediments were determined using calomel and platinum electrode, respectively. Total C and N of the sediment was determined by combustion using a TruSpec^®^ CN analyser (Leco Corp, St Joseph, MI, USA) [22]. Analysis of the overlying water column was undertaken in an identical way to the sediment pore water, except that total dissolved solids were determined by evaporating the samples to dryness at 105 °C.

### 2.3. DNA Extraction and 16S rRNA Gene V4 Amplicon Sequencing

The DNA extraction from 0.25 g of samples of sediments and streamers was done using a QIAGEN DNeasy PowerLyzer PowerSoil Kit (QIAGEN, Hilden, Germany) according to the manufacturer’s protocol. The extracted DNA was quantified using a BR dsDNA Assay kit and Qubit fluorometer (Invitrogen, ThermoFisher Scientific, Altrincham, UK) and PCR-amplified with the 16S rRNA gene V4 region-specific primers forward F515 (5′-GTGBCAGCMGCCGCGGTAA-3′) and reverse R806 (5′-GGACTACHVGGGTWTCTAAT-3′); PCR reactions were performed using MyTaq Red DNA Polymerase (BioLine/Meridian Bioscience, London, UK). DNA bands of approximately 440 bp were agarose gel-purified using a QIAEX II Gel Extraction Kit (QIAGEN, Hilden, Germany). The purified amplicons were then quantified using a Qubit 4.0 Fluorometer (Invitrogen, ThermoFisher Scientific, Altrincham, UK), pooled in equimolar amounts and sequenced using the Illumina MiSeq platform (Illumina, San Diego, CA, USA), applying 500 cycle v2 chemistry (2 × 250 bp paired-end reads), at the Centre for Environmental Biotechnology, Bangor, UK.

### 2.4. Cultivation of Photosynthetic Eukaryotic Organisms

Samples of streamer (5 g) were used for cultivation of eukaryotes (algae). Streamers were washed in 20 mL of the modified DSMZ medium 88 (pH 1.7), previously successfully used for cultivation of acidophiles [13,15,18], containing (g L^−1^): 1.3 (NH_4_)_2_SO_4_, 0.28 KH_2_PO_4_, 0.25 MgSO_4_ ∙ 7H_2_O and 0.07 CaCl_2_ ∙ 2H_2_O. The medium was also supplemented with trace element solution SL10 (DSMZ, https://www.dsmz.de/microorganisms/medium/pdf/DSMZ_Medium320.pdf, accessed on 1 February 2022), added at a 1:1000 (*v*/*v*) ratio, and Kao-Michayluk vitamin solution (Sigma-Aldrich, Gillingham, UK), added at a ratio of 1:100 (*v*/*v*). Washed streamers and suspended materials after the wash-off were then sub-divided and plated using sterile loops onto medium 88 (DSMZ) solidified with a Gelrite (Sigma), 30 (g L^−1^). To generate axenic cultures, plates were supplemented with ampicillin (100 mg L^−1^) to inhibit the bacterial growth. The plates were found to have high levels of fungal contamination; therefore, subsequent samples were grown on plates also supplemented with carbendazim (40 µg mL^−1^). All plates were incubated for 21 days at 20 °C and subjected to photoperiod 12 h:12 h light/dark cycles, with photosynthetic active radiation (PAR) during the light period corresponding to 105 µmol photons m^−2^ s^−1^.

### 2.5. Colony Biomass DNA Extraction

The biomass grown on plates was swabbed using a sterile loop and placed into 1.5 mL Eppendorf tubes for DNA extraction. Cells were lysed using a micropestle in 600 µL of DNA extraction buffer containing 20 mM Tris (pH 8.0), 10 mM EDTA (pH 8.0), 100 mM NaCl and 1% (v/vol) SDS. Samples were incubated at 65 °C for 10 min. After incubation, 150 µL of 5 M potassium acetate was added to the extraction buffer and kept on ice for 10 min. Cell debris were removed by centrifuging samples at 13,000× *g* for 10 min, and the supernatant, containing DNA, was mixed with 100% isopropanol in 1.5 mL microcentrifuge tubes at a 1:1 (vol/vol) ratio. A DNA pellet was obtained by centrifugation at 13,000× *g* for 10 min and washed twice with 70% ethanol, with a 1 min centrifugation step at 13,000× *g* between each wash to re-pellet the DNA. DNA was resuspended in 50 µL DNase free water and stored at −20 °C.

### 2.6. 18S rRNA Gene Sequencing

18S rRNA genes were amplified by PCR with MyTaq DNA Polymerase (Bioline/Meridian Bioscience, London, UK) using universal eukaryotic primers obtained from [23] EukF Forward Primer (5′-AAC CTG GTT GAT CCT GCC AGT-3′) and EukR Reverse Primer (5′-TGA TCC TCC TGC AGG TTC ACC TAC-3′). PCR conditions for the reaction were: 95 °C for 3 min, 35 cycles of 95 °C for 30 s, 60 °C for 15 s, 72 °C for 45 s and a final extension step at 72 °C for 5 min. Successful PCR amplification products of ca. 1 kb and 1.5 kb bands were extracted using the QIAquick Gel Extraction Kit (QIAGEN, Hilden, Germany) and outsourced to Macrogen Ltd. (Amsterdam, The Netherlands) for Sanger sequencing.

### 2.7. Bioinformatic Analysis

Raw sequencing reads were processed according to previously described protocols [13,24]. Briefly, the data were pre-processed to extract the barcodes from sequences and then cleaned of primer sequences using tagcleaner [25]. The barcodes and the sequences were re-matched again using in-house Python scripts. The resulting filtered reads were analysed using QIIME2 v2021.2 [26]. First, the libraries were demultiplexed based on the different barcodes. Then, sequences were corrected and submitted to quality control by the DADA2 pipeline implemented for QIIME2, classifying the reads by amplicon sequencing variant (ASV). Finally, taxonomic assignation of ASVs was performed using SILVA database v138 [27].

### 2.8. Phylogenetic Analysis

In case of Archaea, all ASVs assigned to this domain were included in the phylogenetic tree, whilst in case of Bacteria a selection of a representative sequence for each group was performed based on abundance and length of each ASV of the represented group. Reference sequences were selected after using BLAST+ v2.7.1 [28] with our target sequences against NCBI RefSeq-Rna and NCBInr databases. Overall, the phylogenetic trees were built using 66 and 89 sequences for archaea and bacteria, respectively.

Each set of sequences were aligned using Mafft v7.310 [29] with the L-INS-i algorithm. The resulting multiple alignment was trimmed using trimal v1.4 [30], removing columns with gaps in more than the 20% of the sequences or with similarity scores lower than 0.001. Both archaeal and bacterial final alignments had lengths of 256 bp and were used to build a maximum likelihood phylogenetic tree using the GTR model, with bootstrapping of 1000 pseudoreplicates. Graphical development was performed in the R programming environment [31] using basic tools and the package ape [32].

All statistical analyses were carried out using the R programming environment [31]. Principal component analysis (PCA) was performed using the prcomp function and in-house scripts for graphical design. A multiple regression model was calculated using the lm function, with all metals’ concentrations considered as putative predictor variables. From the initial model, the best possible model was chosen using the function stepAIC from the package MASS [33], selecting the best variables explaining our data for the model. A quantile–quantile plot was calculated to assess the accuracy of the model and to estimate coefficients, and added variable plots were produced to specifically analyse the influence of each selected predictor variable (metal) in the model.

## 3. Results

### 3.1. Environmental Characteristics of the Site

Chemical analysis revealed that the pH of the water and sediment were similar at all sampling points: 2.25 (PM5.1), 2.20 (PM5.2) and 2.19 (PM5.3). The Eh (mV) values of the water and sediment samples were 620 and 526 for PM5.1, 620 and 520 for PM5.2 and 628 and 620 for the site PM5.3, respectively. The pH of the PM4 site was 1.7. The temperature of water and sediment at the time of sampling was 10 °C, while the air temperature was 8.5 °C.

The total carbon and nitrogen values of the sediments were 3.03% and 0.325%, respectively, giving a C:N ratio of 9.37 (Appendix A). The concentrations of total P and inorganic P (mg L^−1^) were 0.1 and 0.75 in the water. Content of phosphate and total P (mg L^−1^) in pore water sediment were measured as 0.54 and 0.11. The total elemental composition in water and sediment, as well as conductivity, total dissolved solids (TDS) and moisture, are shown in Appendix A. Average pH for the three PM5 locations determined in the laboratory was slightly higher in comparison to the measurements at the site (2.44 for water and 2.67 for sediment samples).

The chemical content of sediment and water samples confirmed that the PM5 sites had elevated quantities of metals and metalloids, which was also shown in previous investigations for other Parys Mt sites [19]. As a result of comparative analysis of the chemical composition, we observed certain differences between PM5 and PM4 sites [13,14]. The abundance profile of the different metals showed variability in the two analysed sites (Figure 2). Firstly, the contents of elements such as V and Cr were found to be several orders of magnitude lower in PM5 sediment than in PM4 (Figure 2). Mn was also present in lower amounts in PM5 sediment in comparison to PM4. Sediment concentrations of Al, K, Ca, Fe, Ni, Cu, Zn, Sr, Pb, Rb, As and Y at PM5 were broadly comparable with those at PM4; however, the concentrations of K, Ca, Cu, Zn, Sr and Pb in PM5 water were lower than at the PM4 site. In contrast, S concentrations were much higher at PM5 relative to the PM4 site (12 vs. 1.9 g L^−1^). Of note, P was detected at much higher levels at the PM5 site relative to PM4 [13,14]. The influence of metals on the bacterial relative abundance was determined using multiple linear regression analysis. The best fitting model included V, Mn, Cr, S and As as the best predictor variables (Figure 3).

### 3.2. 16S rRNA Gene Amplicon Sequencing Data

Proportions of sequences from total number of reads derived from archaea in the sediment of the site PM5 were 1.8% (PM5.1), 5.7% (PM5.2) and 16.4% (PM5.3). The most numerous archaeal reads in the site PM5.3 originated from organisms similar to TMEG (12.3% of all ASVs); however, the presence of this group in two other locations in the pool was rather minor (0.5 and 0.7 % in sites PM5.1 and PM5.2, respectively). Other archaea of the class Thermoplasmata were represented by “E-plasma” (the variant of uncultured Thermoplasmatales) and additional archaea of the order Thermoplasmatales (B_DKE or *Ca*. Scheffleriplasma) [34] in rather low quantities in all three locations: 1.2% and 0.08%, 3.7% and 1.4% and 2.8% and 1.4% in PM5.1, PM5.2 and PM5.3 sites, respectively (Figure 4). *Ca*. Micrarchaeota (or *Ca*. Microcaldota) were seen as another minor archaeal group in amounts of 0.2% and 0.03% of the total ASVs in the sites PM5.1 and PM5.3 only (Figure 4, Figure 5 and Figure 6, Appendix A). The diversity of this group was represented by several phylotypes (Figure 6).

Among bacteria, the highest relative abundance was exhibited by Alphaproteobacteria (altogether in quantities 25.4%, 35.6% and 15%) across all three PM5 subsystems, with two genera from the family Acetobacteraceae (order Acetobacterales), *Acidiphilium* (11.9%, 8.6% and 3.7%) and *Acidisphaera* (5%, 19.7% and 5.2%), being the most abundant. The Gammaproteobacteria were detected in 4.7%, 1.1% and 0.08% of total ASVs, with a predominant phylotype being the RCP1-48 group of uncultured Acidithiobacillaceae, order Acidithiobacillales.

In the sediment samples, Proteobacteria were followed by the second most-represented group, uncultured Acidobacteriota (subgroup 1), constituting 20.6, 19.5 and 14.3% of prokaryotic reads in sites PM5.1, PM5.2 and PM5.3, respectively. Furthermore, other bacterial sequences were affiliated with uncultured Chloroflexota (AD3 cluster) 12, 11.9 and 13.6% of ASVs, followed by Nitrospirota—specifically, of the genus *Leptospirillum* (Leptospirillia/Leptospirillales/Leptospirillaceae)—representing 18.6, 4 and 9.8% of ASVs, and uncultured Planctomycetota/CPIa-3 termite group (Phycisphaerae/Tepidisphaerales) and Gemmataceae (Planctomycetes/Gemmatales) representatives (2.8, 7.9 and 12.6%) across three corresponding locations of the pool PM5 (Figure 4, Figure 5 and Figure 7; Appendix A).

Another well-represented bacterial phylum was Actinobacteriota, with the IMCC26256 group of sequences (about 5% in all locations), and other uncultured Actinobacteriota (9.2%, 6.7% and 10.4%) in sites PM5.1–PM5.3, respectively. Of note, Firmicutes (Bacillota), mostly uncultured Sulfobacillia/Sulfobacillales/Sulfobacillaceae at 2.2, 1 and 1.6 % of ASVs, were minor groups in all sampling spots of the PM5 pool. Small relative numbers of Dependentiae, Cyanobacteria, Bacteroidota, Elusimicrobiota, Myxococcota and “Patescibacteria” were revealed in all three studied locations of the pool as well (Figure 4, Figure 5 and Figure 7; Appendix A).

Additionally, we analysed the affiliation of organisms associated with the filamentous streamer in samples PM5.1 and PM5.2 and found ASVs related to *Acidiphilium* in significant numbers of reads (44 and 68%), with *Leptospirillum* (40 and 5.4%) being the highest, followed by less represented groups: *Acidisphaera* (3.9 and 5.9%), Cyanobacteria (5.7 and 0.7%), and *Acidithiobacillus* (1.3 and 2.2%) in both locations. Furthermore, sequences clustering with uncultured groups within Actinobacteriota (IMCC26256), Acidimicrobiia, Acidobacteriaceae/subgroup 1, Planctomycetota/Gemmataceae, Chloroflexota (AD3 cluster) and others were found as minor components associated with the PM5 streamer (Appendix A). No archaea were detected in ASVs derived from the streamer growth.

### 3.3. Identification of Phototrophic Eukaryotes Associated with Streamer

Single green pigmented colonies of three types were observed on DSMZ medium 88 plates after 21 days of Parys Mt streamer sample cultivation in the light. The first colony type was observed in a light microscope as small (~8 µM), oval- to round-shaped cells. 18S rRNA gene sequencing revealed a 99% similarity to green phytoflagellate *Chlamydomonas acidophila* (AJ783841) (Viridiplantae/Chlorophyta/Chlorophyceae/Chlamydomonodales). The cells of a second species were observed by microscopy as long, worm-like cells (~30–50 µM long), and 18S rRNA gene sequencing revealed a 98% sequence similarity to protists *Euglena* cf. *mutabilis* (AY082988) (Euglenozoa/Euglenida/Euglenales/Euglenaceae) isolated from the acidic river Rio Tinto, Spain. The third colony type was identified as the moss *Bryopsida* sp., sharing a 97% sequence similarity to *Bryopsida* sp. RSa4 (KM016994) (Viridiplantae/Streptophyta/Bryophyta) isolated from an acidic stream draining a copper mine tailing in Serbia (Figure 8).

## 4. Discussion

The prokaryotic microbiome of the PM5 site exhibited a much lower abundance of archaea in comparison to the PM4 site [13,14]. Such a low abundance of archaea in the PM5 site may be explained by a number of factors. The first aspect is the higher pH value of PM5 (pH 2.20) relative to PM4 (pH 1.7). Our result is in accordance with previous studies showing that pH is consistently considered as the main factor shaping microbial community in sulphidic mine tailings [35,36]. Archaea, bacteria and eukaryotes populating low pH environments employ numerous mechanisms of adaptation, with archaea reported as the most acidophilic life forms [37].

In addition, dissolved solutes at PM5 (3.2–3.9 mS cm^−1^) were significantly lower in comparison to PM4 (7.4–10.3 mS cm^−1^) [13] and only slightly higher than the mean value for the Dyffryn Adda water system on Parys Mt, studied earlier in [7]. Higher dissolved solutes and lower pH were also found previously to favour the presence of archaea at another north Wales, UK, mine located at Cae Coch [11].

It is also likely that carbon and nitrogen availability might strongly influence the structure of the microbial community and relative numbers of archaea and bacteria. However, the values of carbon and nitrogen concentrations in the sediments were in the same range for both PM4 and PM5 sites [13,14]. Possibly, the organic carbon exuded into the water from the extensively growing primary producers (e.g., *Chlamydomonas acidophila*, *Euglena mutabilis* and *Bryopsida* sp.) also favours bacterial growth, or their secondary metabolites could be toxic for archaea. Whatever the case, the availability and nature of organic carbon in PM5 and PM4 might be an important factor shaping the community of heterotrophic prokaryotes. Certain differences in chemical content were observed between PM5 and PM4 sites [13,14]. The concentrations of V and Cr were a few orders of magnitude lower in PM5 sediment than in PM4. Sulphur was more abundant in PM5 relative to the PM4 site. Overall, it appears that a range of interacting biotic and abiotic factors probably shape the content of the microbial prokaryotic acidophilic community in the PM5 site, leading to domination by bacteria.

In terms of taxonomic diversity, it should be noted that the archaea in the PM5 site were represented by ASVs, similar to that in the site PM4, although in insignificant relative numbers, suggesting that their input into the community metabolism and element cycling is minor compared to bacteria. The most abundant archaeal group across all Parys Mt sites were Thermoplasmatales with the “E-plasma” variant, which was a major taxonomic group in the site PM4 (58% of all prokaryotic metagenomic shotgun and ASV reads) [13]. This emphasizes that the “E-plasma” phylotype might be the most successful archaeal coloniser of Parys Mt. Further Thermoplasmatales archaea were affiliated with additional uncultured organisms represented by accession numbers KF225694, EU370308 and KJ907755, detected at Los Rueldos AMD system, Rio Tinto (both Spain) and Los Azufres (Mexico). Furthermore, “B_DKE” and other Thermoplasmatales archaea, all of which were detected in the site PM4, were shown to be present in minor numbers in the site PM5 [13]. Moreover, our cultivation experiments revealed the presence of *Cuniculiplasma* and *Ferroplasma* that were not detected in situ in PM5 by 16S rRNA gene amplicon sequencing [38]. Altogether, this suggests the dominance of organotrophic and probable iron-oxidising metabolic activity of archaea at the PM5 site [39].

The relative numbers of TMEG-associated sequences in PM5 were rather comparable with those at PM4, yet the highest relative numbers of this archaeal group were identified in the PM5.3 samples, the most acidic and oxic location of the PM5 pool. It was previously shown that these archaea are distributed in different environments, and sequences of TMEG representatives inhabiting low pH settings were shown clustering together and forming a separate order within the class Thermoplasmata [13]. Further, the sequences of TMEG found in the PM5 site were represented by two phylotypes similar to organisms represented by accession numbers FN862155 and KF225676 found at the Spanish Rio Tinto and Los Rueldos AMD sites, respectively [40,41]. Sequences clustering with this group were also detected in another Parys Mt study [7], emphasizing the ubiquity and the diversity of phylotypes of these organisms in Parys Mt ecosystems. There are, however, no records on any cultivation success for representatives of this group. We need to note that our efforts to enrich for, and cultivate, the TMEG were also unsuccessful.

Some variability was also observed in ASVs affiliated with *Ca*. Micrarchaeota in PM5. Three distinct phylotypes were identified, one of which was distantly related to *Ca*. Mancarchaeum acidiphilum (91% of sequence identity), and another was clustered with the further micrarchaeotal reads from the site PM4 [15,42]. The third group of *Ca*. Micrarchaeota ASVs were identical with sequences GQ141778 and JF280346 from volcanic environments in Costa Rica and Colombia, correspondingly. Thus, our data suggest that Thermoplasmatales, TMEG and *Ca*. Micrarchaeota are typical indigenous archaeal taxa present across the Parys Mt sites.

Bacterial taxa found herein were either well-known, highly abundant inhabitants of, or organisms previously detected in, AMD environments with confirmed organotrophic and/or iron- and sulphur-metabolising capabilities [43]. *Acidiphilium*, *Acidisphaera*, *Leptospirillum* and *Acidithiobacillus* were previously identified in the Parys Mt underground lake [10]. Proteobacteria and Nitrospirota were found to be the major groups in both PM sites (PM4 and PM5) and are generally known to be abundant prokaryotes in AMD settings, contributing to carbon, nitrogen, iron and sulphur cycling; however, we observed the prevalence of other bacterial taxa in contrast to PM4 water and sediment [13,14,37]. Firstly, in the site PM5, the members of the classes Acidobacteriota, Chloroflexota and Planctomycetota were highly abundant. The majority of these organisms are yet uncultured, with largely unknown metabolic potential and roles in the environment. Further “microbial dark matter” representatives—e.g., IMCC26256 (Acidimicrobiia), AD3 (Chloroflexota), RCP2-54 (Desulfobacterota) and, in lower quantities, RCP1-48 (Proteobacteria) and lineage IV of “Elusimicrobiota”—are worth noting.

The biodiversity of the streamer-associated prokaryotic community revealed significant levels of associated Proteobacteria (mostly as sequences of heterotrophic iron-reducing genus *Acidiphilium*) and Nitrospirota (as the iron-oxidizing and metal sulphide-oxidising genus *Leptospirillum*), with the streamer pointing at their massive accumulation in this particular environment. It should be added that microalgae in AMD systems are widely distributed and studied [44,45]. Acid streamers and gelatinous filamentous growth are widely distributed in AMD environments and host iron- and sulphur-oxidising phototrophic, chemoautotrophic and heterotrophic microorganisms [46,47,48,49]. In particular, *Acidiphilium* spp. were found in associations with macroscopic growth and were previously assumed to be supported by algal exudates [50,51]. Heterotrophic *Acidiphilium* are known players in mine waters and were earlier isolated from Parys Mt sites [16]. A previous study at Parys Mt [20] also identified the role of microalgae in sustaining the metabolic needs of heterotrophic acidophilic bacteria of genera *Acidiphilium*, *Acidobacterium* and *Acidocella* via the consumption of monosaccharides. *Leptospirillum* spp. were also detected to be dominant organisms inhabiting acid streamers and slimes [46]. Furthermore, *Leptospirillum* were documented to encode quorum sensing with c-di-GMP pathways [52]. A relatively constant temperature regime in these sites may be another factor worth considering in relation to the microbial community composition; although, e.g., *Leptospirillum* spp. were previously documented to favour more temperate conditions [7]. Of note, streamer-forming bacteria were isolated from Parys Mt and from another acidic environment of north Wales and described as *Ferrovum myxofaciens* [8]. However, we did not observe filamentous growth in heterotrophic enrichment cultures established with Parys Mt samples [38] and no *Ferrovum-* affiliated reads were identified in the streamer community, either.

Archaea were not found to be associated with macroscopic filamentous growth in this study. In relation to this, the absence of archaea within the streamer points at their other niche preferences in Parys Mt—namely, sediments—which was demonstrated previously [13,14]. However, archaea, and particularly *Ferroplasma* spp., were found occupying these microniches: acid streamers and other macroscopic growth formations [46,50]. Whatever the reason for the archaeal scarcity in streamers, it should be noted that *Chlamydomonas acidophila* favoured extreme conditions in different settings on Parys Mt [19,53]. Other eukaryotes associated with the streamer growth were *Euglena mutabilis* and *Bryopsida* sp., previously shown to be associated with mine acidic streams [54,55].

## 5. Conclusions

The dominance of bacteria over archaea in microbial communities of the site PM5 seems to be primarily linked with the higher pH values and reduced dissolved solute levels being the most determining parameters. The sediment was identified as a preferable niche for archaea in this environment. Bacteria (but not archaea) were found to be associated with eukaryotic streamer growth, with a clear predominance of the two particular genera *Acidiphilium* and *Leptospirillum*. Predominance of *Acidiphilium* and presence of other heterotrophic bacteria could be explained by access to organic substrates provided by eukaryotes. Taxonomic composition and predicted metabolic potential (heterotrophy and iron cycling) might be considered as determined by similar archaeal and bacterial groups across all currently studied Parys Mt settings, pointing, however, to some environmental preferences for these organisms at the level of microniches and dependence on the geochemistry of acidic formations represented in this environment.

## Figures and Tables

**Figure 1 microorganisms-10-01694-f001:**
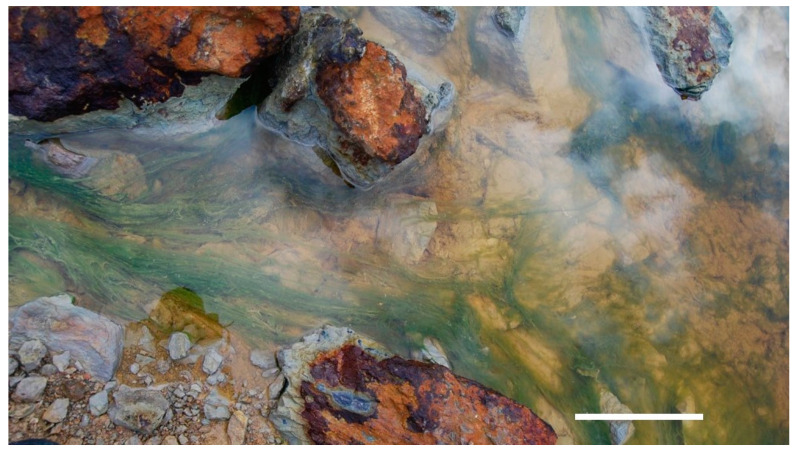
Macroscopic eukaryotic filamentous growth in the sampling site PM5, Parys Mt, UK. Scale bar, 0.1 m.

**Figure 2 microorganisms-10-01694-f002:**
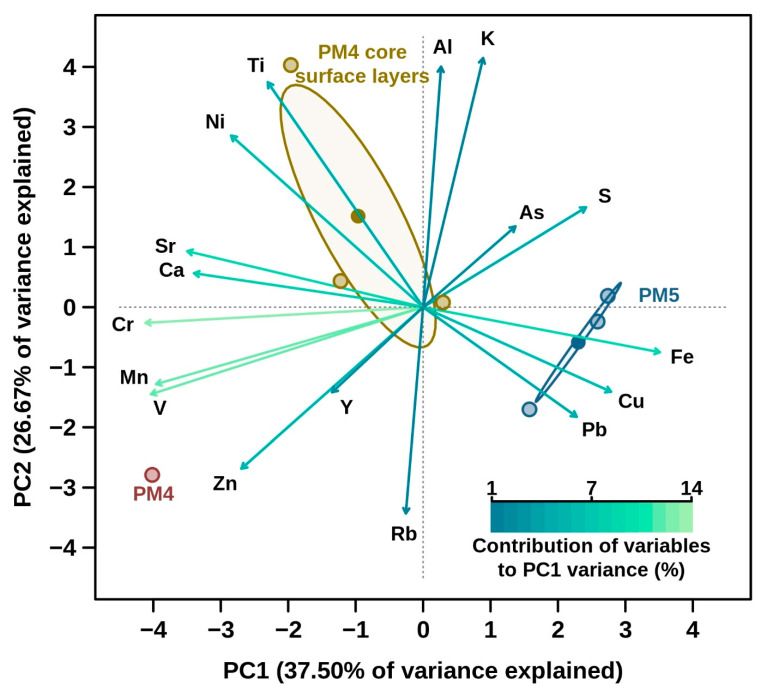
Principal component analysis (PCA) representing the abundance profiles of different metals present in the two analysed Parys Mt sites (PM4, red circles; PM5 blue circles), including, additionally, samples from the core surface layers (yellow circles) collected from PM4 sediments [13]. Ellipses are drawn for PM4 surface core samples (yellow) and for PM5 samples (blue) based on the samples’ variance. Corresponding dots are based on samples’ means. The contribution (%) of each metal to samples’ separation over PC1 (X axis) is shown by the arrows (colour key is shown in the bottom right).

**Figure 3 microorganisms-10-01694-f003:**
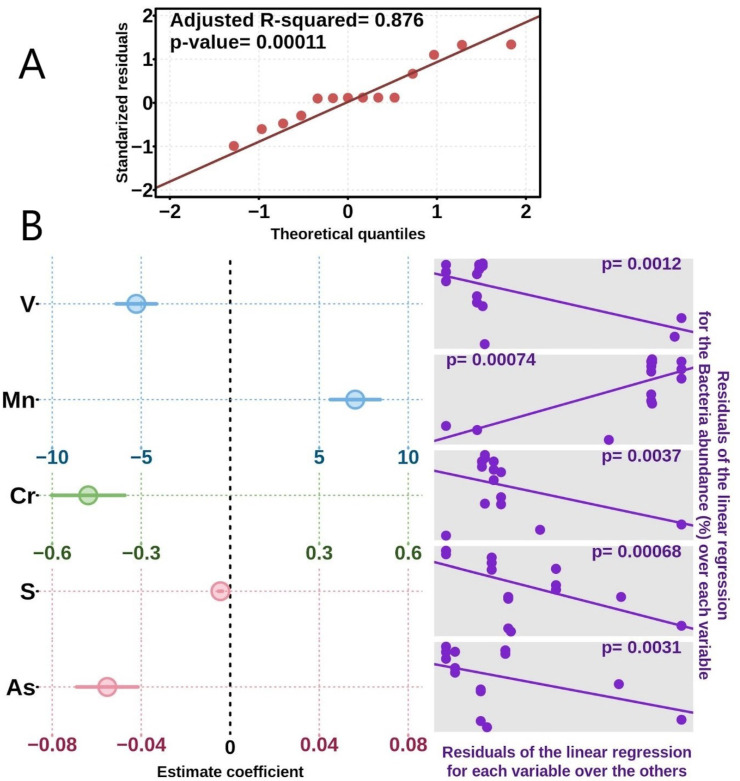
Analysis of the influence of metals on the bacterial relative abundance with multiple linear regression. The best-fitting model included V, Mn, Cr, S and As as the best predictor variables. (**A**) Quantile–quantile plot showing the accuracy of the best obtained model, after analysing all metals profiles. (**B**) The estimate coefficient of each predictor variable (metal) is shown on the left. Each colour represents a different x-axis scale. On the right side of each variable, the corresponding variable plot is shown.

**Figure 4 microorganisms-10-01694-f004:**
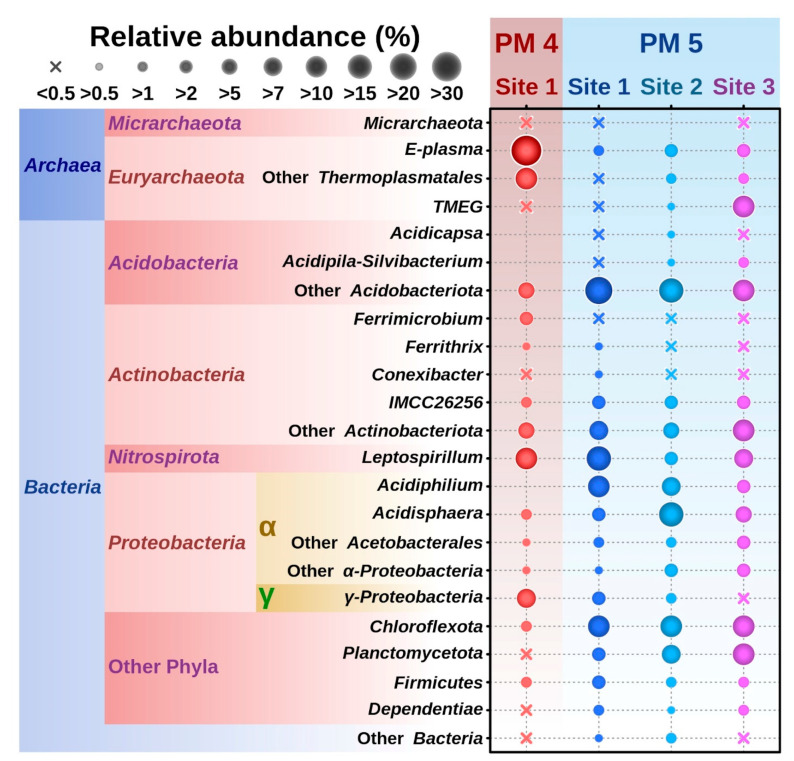
Relative abundances of 16S rRNA gene amplicon reads of the prokaryotic most represented taxonomic groups (MATGs) across the three localities of the PM5 site (PM5.1, PM5.2 and PM5.3) and PM4 site, Parys Mt. Circles represent the average relative abundance (%), *n* = 3. MATGs were selected according to a > 2% cut off, except for *Ca*. Micrarchaeota.

**Figure 5 microorganisms-10-01694-f005:**
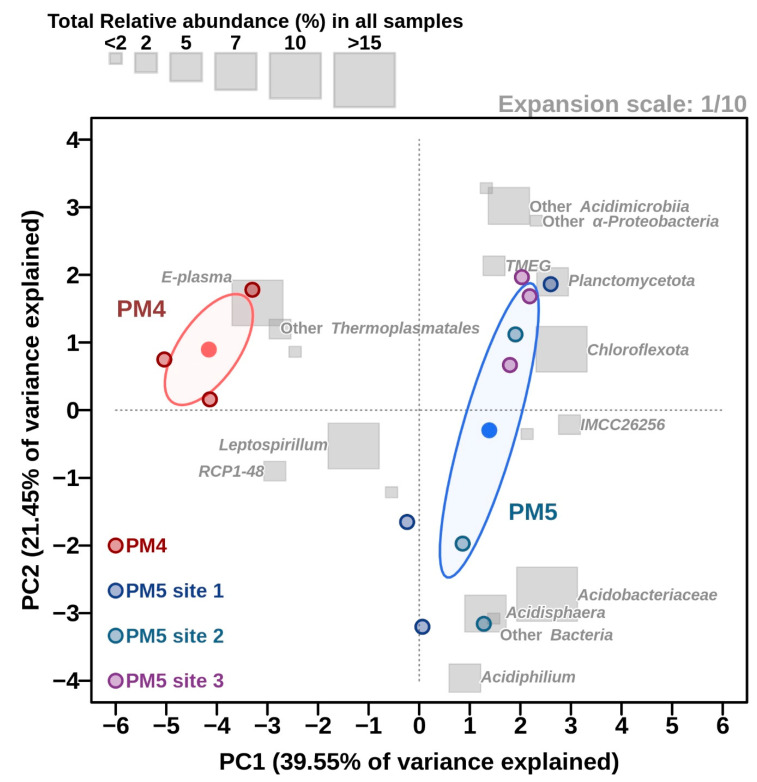
Principal component analysis (PCA) for the relative read numbers of the most-abundant taxonomic groups (MATGs). Any taxon is considered a MATG if it reaches a relative abundance over ≥2%; otherwise its relative read counts are added to the immediate upper lineage level. Once all genera were sorted, the upper level of the lineage was analysed in the same way but without considering the corresponding relative abundance already covered in the MATGs. Ellipses were calculated for the two different locations (PM4 in red and PM5 in blue), based on the mean (dots at the ellipse centre) and the variance among the samples. Total relative abundance of each MATG among all samples is shown by the grey rectangles. The expansion scale represents the proportion at which variables’ (MATGs’) axes are drawn regarding the site axes.

**Figure 6 microorganisms-10-01694-f006:**
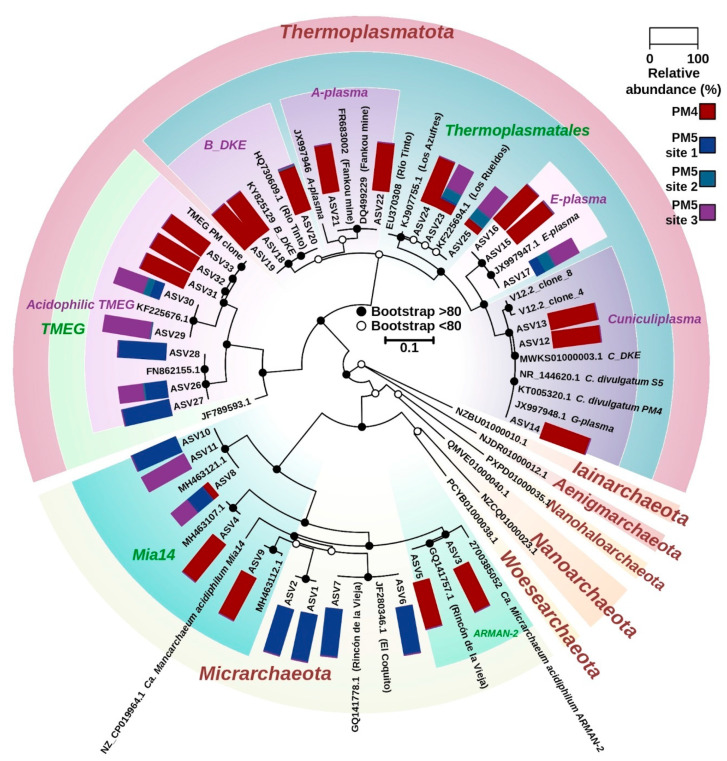
Phylogenetic maximum likelihood tree based on 16S rRNA gene sequences of archaea from sites PM4 and PM5 (PM5.1, PM5.2 and PM5.3), Parys Mt. Relative abundance (%) of reads assigned to each ASV found on PM4 and PM5 sites is shown in the rectangles attached to each ASV label. Bootstrap score is shown by open (<80) or filled (>80) circles for each node. Scale bar, 0.1 substitutions per position.

**Figure 7 microorganisms-10-01694-f007:**
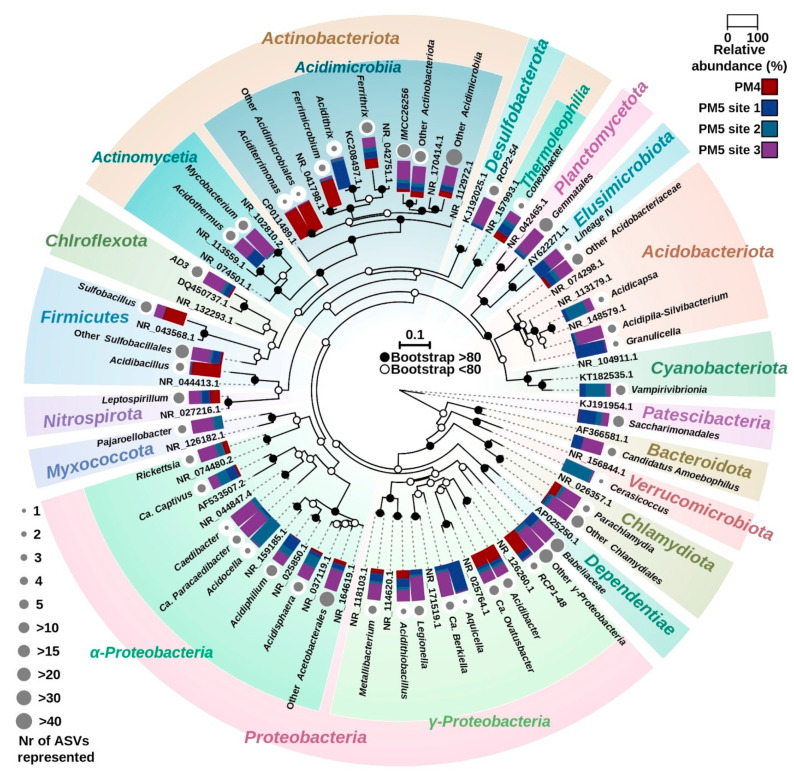
Maximum likelihood phylogenetic tree based on 16S rRNA gene sequences of bacteria from sites PM4 and PM5 (PM5.1, PM5.2 and PM5.3), Parys Mt. Relative abundance (%) of reads assigned to taxonomic groups found on each site are shown in the rectangles attached to respective groups. Taxonomic groups correspond to the sum of all ASVs by lineage and make groups, taking one ASV sequences as representative and highlighting the number of ASVs in the groups through the size of the circle next to it on the tree. The number of ASVs assigned to each represented group is shown by the size of the grey circle next to each label. Bootstrap score is shown by open (<80) or filled (>80) circles for each node. Scale bar, 0.1 substitutions per position.

**Figure 8 microorganisms-10-01694-f008:**

Microscopic images taken from axenic colonies isolated from PM5, grown on Gelrite plates with DSMZ medium 88. (**A**) Round- and oval-shaped cells of *Chlamydomonas acidophila* under 100× magnification; (**B**) a single cell of the protist *Euglena mutabilis* under 100× magnification; (**C**) *Bryopsida* sp. under a dissecting microscope at 5× magnification.

## Data Availability

The nucleotide sequence data identified in this study are available under BioProject PRJNA850927 and accession number ON831504-6 (eukaryotic sequences).

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
