# Peer review of "Microbial Diversity of a Disused Copper Mine Site (Parys Mountain, UK), Dominated by Intensive Eukaryotic Filamentous Growth"

_microorganisms, 2022, doi:10.3390/microorganisms10091694_

Round 1

Reviewer 1 Report

The subject of the manuscript is consistent with the scope of the Journal. The topic of research is interesting. The present paper is prepared in the usual manner for scientific work, both the division into chapters collected results in the form of figures and tables.  The organization of the manuscript is satisfactory, and the length of manuscript is appropriate to the content.

I agree with the authors that research of this type is important.

Please, be sure that all the references cited in the manuscript are also included in the reference list and vice versa with matching spellings and dates.

Author Response

The subject of the manuscript is consistent with the scope of the Journal. The topic of research is interesting. The present paper is prepared in the usual manner for scientific work, both the division into chapters collected results in the form of figures and tables.  The organization of the manuscript is satisfactory, and the length of manuscript is appropriate to the content.

I agree with the authors that research of this type is important.

 Please, be sure that all the references cited in the manuscript are also included in the reference list and vice versa with matching spellings and dates.

We thank the Reviewer for their positive assessment, all references have  additionally been checked.

Reviewer 2 Report

In general, the introduction presents the essential contents to including the hypothesis of the work, however, it is not fluent to read and between the first part (up to line 45) and the second part there does not seem to be a connection, I suggest to review the draft carefully of this part. Even the materials and methods are not very fluent and in some places, there are no clarifications (see below my comments line by line) but in general, it is quite clear how the authors completed the work. Results are instead described in an exhaustive manner, and the inserted graphs are clear and easy to understand, even if some changes are made to figure 1 (see my comments on the figures) I would also like figure S2 to be included in the work as it is explanatory and useful about the part of identifying eukaryotes. Discussions and conclusions are clear enough as to what has been achieved.

Comments line by line

Line 98: "a yellowish-green biofilm, which 98 grows and fills the pool from the surface to the bottom" should be indicator first.

Line 129: In any case, the sequences of the primers used should be inserted

Line 134: how were the samples sown? Has a preliminary treatment or dilutions been carried out? to specify.

Line 187: Indicating the analyzes carried out.

Line 284: AB? in the materials and methods the culture medium 88 is indicated to clarify this discrepancy.

Lines 287-288: These lines would go into materials and methods.

Figures

Figure 1: In the figure, it is not very clear where the bacteria separate from the archaea, perhaps the wording "Bacteria" could be moved

Author Response

In general, the introduction presents the essential contents to including the hypothesis of the work, however, it is not fluent to read and between the first part (up to line 45) and the second part there does not seem to be a connection, I suggest to review the draft carefully of this part.

The sentence Finally, the site is an industrial archaeological monument of international importance being designated as a World Heritage Site due to the rich history of exploitation which started in the Bronze Age was removed and we hope that the first part (up to line 45) and the second part are connected now.

Even the materials and methods are not very fluent and in some places, there are no clarifications (see below my comments line by line) but in general, it is quite clear how the authors completed the work. Results are instead described in an exhaustive manner, and the inserted graphs are clear and easy to understand, even if some changes are made to figure 1 (see my comments on the figures) I would also like figure S2 to be included in the work as it is explanatory and useful about the part of identifying eukaryotes.

Changed, the Fig.S2 is Fig.8 now.

Discussions and conclusions are clear enough as to what has been achieved.

Comments line by line

Line 98: "a yellowish-green biofilm, which 98 grows and fills the pool from the surface to the bottom" should be indicator first.

Changed to: yellowish-green biofilm (filamentous acid streamer), growing in, and filling the pool from the surface to the bottom

Line 129: In any case, the sequences of the primers used should be inserted

Done: The DNA extraction from 0.25 g of samples of sediments and streamers was done using a QIAGEN DNeasy PowerLyzer PowerSoil Kit (QIAGEN, Hilden, Germany) according to the manufacturer’s protocol. The DNA was quantified using BR dsDNA Assay kit and Qubit fluorometer (Invitrogen, Carlsbad, CA, UK) and PCR-amplified with the 16S rRNA gene V4 region-specific primers: forward F515 (5’-GTGBCAGCMGCCGCGGTAA-3’) and reverse R806 (5’-GGACTACHVGGGTWTCTAAT-3’) following the same procedures as specified earlier by [13].

Line 134: how were the samples sown? Has a preliminary treatment or dilutions been carried out? to specify.

The text now reads: Samples of streamer (5 g) were used for cultivation of eukaryotes. Streamers were washed in 20 ml of the modified medium 88 (DSMZ) (pH 1.7) containing (g L-1): 1.3 (NH4)2SO4, 0.28 KH2PO4, 0.25 MgSO4 ∙ 7 H2O, 0.07 CaCl2 ∙2 H2O. The medium was also supplemented with trace element solution SL10 (DSMZ, https://www.dsmz.de/microorganisms/medium/pdf/DSMZ_Medium320.pdf) added at a 1:1000 (v/v) ratio and Kao-Michayluk vitamin solution (Sigma-Aldrich, UK) at a ratio of 1:100 (v/v). Washed streamers and suspended materials after the wash-off were then sub-divided and plated using sterile loops on to medium 88 (DSMZ) solidified with a Gelrite (Sigma), 30 (g L-1). To generate axenic cultures, plates were supplemented with ampicillin (100 mg L-1) to inhibit the  bacterial growth. The plates were found to have high levels of fungal contamination, therefore subsequent samples were grown on plates also supplemented with carbendazim (40 µg mL-1). All plates were grown at 20 °C and subjected to 12 h:12 h light/dark cycles.

Line 187: Indicating the analyzes carried out.

We have added the following text All statistical analyses have been carried out using R programming environment [31]. Principal Component Analysis (PCA) has been performed using prcomp function and in-house scripts for graphical design. Multiple regression model was calculated using lm function, with all metal’s concentrations considered as putative predictor variables. From the initial model, the best possible model was chosen using the function stepAIC from the packages MASS [33], selecting the best variables explaining our data for the model. Quantile-Quantile plot was calculated to assess the accuracy of the model and to estimate coefficient and added variable plots were done to analyse specifically the influence of each selected predictor variable (metal) in the model.

Line 284: AB? in the materials and methods the culture medium 88 is indicated to clarify this discrepancy.

The text has changed for: Single green pigmented colonies of three types were observed on medium 88 (DSMZ) plates after 21 days of Parys Mt streamer sample cultivation in the light.

Lines 287-288: These lines would go into materials and methods.

Done.

Figures

Figure 1: In the figure, it is not very clear where the bacteria separate from the archaea, perhaps the wording "Bacteria" could be moved

Done. The Fig.1 is Fig. 3 now.

Reviewer 3 Report

The paper is devoted to microbial diversity of the sediments of disused copper mine in Wales, UK. The site is ecologically interesting and important due to its specific location and highly acidic pH.  The methodology used is appropriate for the study purposes. However, there are some aspects, which need to be clarified and improved.

The title mentions filamentous growth but it should specify of which kind, because filamentous growth can be also of fungal nature.

Abstract

Hereafter no taxonomic names of the ranks higher than genus should be written in italic.

Materials and Methods

The section should start with the subsection devoted to site description (climate, soil, geology, etc.)

2.9. Statistical Analysis – should contain the brief description of the analyses performed.

Results

Lines 195-196: The reference to this figure should be placed in the subsection 2.1., but not here.

It is unclear for me why not to combine the subsections 3.1.and 3.2 in one subsection, because here only the temperature measurements do not belong to chemical analyses. While the pH values are given for each locality studied, the total carbon and nitrogen values are presented only in average. Why? Are the values in each locality not important for further analysis? And why is pH for PM.4 not given?

Lines 205-217: It is hard to follow the information in this paragraph, and the percentages unnecessarily given up to hundredths create an additional complication.

Figure 5. It is confusing to mix the groups of different taxonomic ranks - genera, families, and phyla, in taxonomic affiliation.

Discussion

Lines 304-307: on my mind, it is rather an insufficient explanation. Both pH values are extremely acidic, and it is unclear whether the differences between them may really explain, even partly, the differences in Archaea abundance.

Lines 320-332:  This information should be placed in Results, but not in Discussion, and Fig. 6 should be placed before the figures showing the composition of microbial communities.

Overall, the authors use the measured abiotic parameters only for the approximate explanations of the variations in composition of microbial communities and do not support them by any statistical analyses showing the relationship between biotic and abiotic characteristics, like linear regression, multiple regression, RDA, etc. Such analyses would definitely strengthen the explanations and the conclusions. I think also that the authors should more deeply discuss the influence of extremely low pH of the sediments on the composition of prokaryotic and cultivated eukaryotic communities. Additionally, the section contains some information like accession numbers of sequences, which is unnecessary for Discussion.

All other suggestions and corrections are inserted into the PDF version of manuscript, which is attached.

Author Response

The paper is devoted to microbial diversity of the sediments of disused copper mine in Wales, UK. The site is ecologically interesting and important due to its specific location and highly acidic pH.  The methodology used is appropriate for the study purposes. However, there are some aspects, which need to be clarified and improved.

The title mentions filamentous growth but it should specify of which kind, because filamentous growth can be also of fungal nature.

We humbly disagree, as the filamentous growths are not formed by a single organism, but are a product of a variety of pro- and eukaryotic taxa, with the latter being :

Chlamydomonas acidophila (Viridiplantae/Chlorophyta/Chlorophyceae/Chlamydomonodales),

Euglena cf. mutabilis (Euglenozoa/Euglenida/ Euglenales/Euglenaceae) and

Bryopsida (Viridiplantae/Streptophyta/Bryophyta).

Abstract

Hereafter no taxonomic names of the ranks higher than genus should be written in italic.

For the ‘Microorganisms’ all taxonomic names on all levels should be written in Italics.

Materials and Methods

The section should start with the subsection devoted to site description (climate, soil, geology, etc.)

The geology of the site with references is specified in the Introduction (please see below).

Thirdly, the area possesses a unique geology, being the only example of Kuroko type massive volcanic-associated sulfide deposits in the UK as well as containing an abundance of the ore deposit anglesite (PbSO4), which was discovered at the site, alongside a range of other Pb, Zn and Fe containing minerals (e.g. pyrite, chalcopyrite, sphalerite, galena). Combined, these aspects make Parys Mt a place of great scientific interest for geochemical and extremophile studies [3-5].

This information was added into the Site description and Sampling section: Parys Mt is the abandoned copper mine (53.38747; -4.33968), on the Island of Anglesey, north-west Wales, UK, a region with a maritime climate.

Soils of the site were out of the focus of this study and are not discussed here.

2.9. Statistical Analysis – should contain the brief description of the analyses performed.

Following text was added: All statistical analyses have been carried out using R programming environment [31]. Principal Component Analysis (PCA) has been performed using prcomp function and in-house scripts for graphical design. Multiple regression model was calculated using lm function, with all metal’s concentrations considered as putative predictor variables. From the initial model, the best possible model was chosen using the function stepAIC from the packages MASS [33], selecting the best variables explaining our data for the model. Quantile-Quantile plot was calculated to assess the accuracy of the model and to estimate coefficient and added variable plots were done to analyse specifically the influence of each selected predictor variable (metal) in the model.

Results

Lines 195-196: The reference to this figure should be placed in the subsection 2.1., but not here.

The reference to this figure in the Results has been removed.

It is unclear for me why not to combine the subsections 3.1.and 3.2 in one subsection, because here only the temperature measurements do not belong to chemical analyses. While the pH values are given for each locality studied, the total carbon and nitrogen values are presented only in average. Why? Are the values in each locality not important for further analysis?

The subsections 3.1 and 3.2 are now combined. The values of the total carbon and nitrogen are important characteristics for acidophiles, but were found in relatively similar values in comparison to PM4 site and were measured as an average for the PM5 site.

And why is pH for PM.4 not given?

Added : The pH of the site PM4 was 1.7.

Lines 205-217: It is hard to follow the information in this paragraph, and the percentages unnecessarily given up to hundredths create an additional complication.

Changed: Percentages are shown up to tenths now. We hope that it is easer to follow the information now.

Figure 5. It is confusing to mix the groups of different taxonomic ranks - genera, families, and phyla, in taxonomic affiliation.

The families and phyla are shown only for cases where the affiliation on the level of genera was not possible.

Discussion

Lines 304-307: on my mind, it is rather an insufficient explanation. Both pH values are extremely acidic, and it is unclear whether the differences between them may really explain, even partly, the differences in Archaea abundance.

With all due respect, the difference between pH 1.7 and pH 2.2 (considering that pH is a logarithmic scale) is significant: corresponding concentrations of protons are 0.02 and 0.0063 mol [H+] /L, i.e. a threefold difference. Acidophilic Archaea, in comparison to acidophilic bacteria, are known to be the most hyperacidophilic of all currently known life forms, able to grow at pH<0 as Picrophilus for example, and this means that Picrophilus is able to grow in solutions with hydronium ion (H3O) concentration up to 100 times greater than other extreme acidophiles (Schleper et al., 1995).

Lines 320-332:  This information should be placed in Results, but not in Discussion, and Fig. 6 should be placed before the figures showing the composition of microbial communities.

The text is moved now to the Results Section, the Fig.6 is a Fig.3 now.

Overall, the authors use the measured abiotic parameters only for the approximate explanations of the variations in composition of microbial communities and do not support them by any statistical analyses showing the relationship between biotic and abiotic characteristics, like linear regression, multiple regression, RDA, etc. Such analyses would definitely strengthen the explanations and the conclusions.

Added: New figure (Fig. 2) with the Multiple Regression analysis made for metal concentration according to the recommendation of the reviewer. However, in line with previous reports we consider that main factors favouring shift in archaea:bacteria numbers for acidophiles assumed to be pH and conductivity.

I think also that the authors should more deeply discuss the influence of extremely low pH of the sediments on the composition of prokaryotic and cultivated eukaryotic communities.

The phrase is added in the context of the influence of extremely low pH on prokaryotes and eukaryotes: Archaea, bacteria and eukaryotes populating low pH environments employ numerous mechanisms of adaptation, with archaea reported as the most acidophilic life forms [37].

Additionally, the section contains some information like accession numbers of sequences, which is unnecessary for Discussion.

Accession numbers give necessary and quite often the only available data about uncultured organisms, and we believe that it would contribute to the factual information represented in the Discussion.

All other suggestions and corrections are inserted into the PDF version of manuscript, which is attached.

Please see here the answers to comments in the text here:

L.20, 21, 23, 30 Latin names of higher taxa than genera. S. above please.

L.59 Another study identified structurally and biogeochemically distinct microbial communities in the upper sediment layers (0-20 cm in depth) with a significant proportion of the community represented by archaea [14].

structurally and biogeochemically (What does it mean?)

Changed to: Another study identified distinct microbial communities in the upper sediment layers (0-20 cm in depth) with a significant proportion of the community represented by archaea [14].

L.74 Phylum is changed to phylum

L.88 The prokaryotic community of PM5 was then compared to the community inhabiting the Parys Mt stream (PM4), which was recently analysed using the same experimental approaches, and which showed archaea to be prevalent.

The comment: why “experimental”, if no experiments were conducted during the study?

Changed to: The prokaryotic community of PM5 was then compared to the community inhabiting the Parys Mt stream (PM4), which was recently analysed using the same methods, and which showed archaea to be prevalent.

L.92 Site description

Please see our comments to this point before.

L.109 Temperature, pH and Eh (the abbreviation should be specified) were measured in the field using SevenGo® multimeter (Mettler-Toledo, Leicester, UK) in triplicate (in each location?).

Done: Temperature, pH and Eh (Redox potential) were measured in the field using SevenGo® multimeter (Mettler-Toledo, Leicester, UK) in triplicate in each location.

L.117 Pore water phosphate (why namely phosphate?) was determined using the colorimetric molybdate blue method of [21].

Phosphate was the only element determined colorimetrically.

L.118 standard electrode, what does it mean?

Changed to: pH and electrical conductivity (EC) of the sediments were determined using calomel and platinum electrode, respectively.

L.118 what is the difference between this pH and pH mentioned at the end of previous subsection?

Added: Average pH for PM5 three locations determined at laboratory was slightly higher in comparison to the measurements at the site (2.44 for water and 2.67 for sediment samples).

L.120 reference for the method

Added: Nelson, D.W., Sommers, L.E. (1996) Total carbon, organic carbon, and organic matter. In: Sparks, D.L. (ed) Methods of soil analysis-part 3: chemical methods. Soil Science Society of America, Madison, WI, pp 961-1010.

L.120-121 unclear

Changed to: Analysis of the overlying water column was undertaken in an identical way to the sediment pore water

L.127 The DNA extraction from 0.25 g of samples of sediments and streamers was done using a QIAGEN DNeasy PowerLyzer PowerSoil Kit (QIAGEN, Hilden, Germany) according to the manufacturer’s protocol, quantified using BR dsDNA Assay kit and Qubit fluorometer (Invitrogen, Carlsbad, CA, UK) and PCR-amplified with the 16S rRNA gene V4 region-specific primers following the same procedures as specified earlier by [13].

The comment: it is unclear what was quantified and PCR-amplified – the extracted DNA? The sentence is too long and it is hard to follow it.

The sentence is modified: The DNA extraction from 0.25 g of samples of sediments and streamers was done using a QIAGEN DNeasy PowerLyzer PowerSoil Kit (QIAGEN, Hilden, Germany) according to the manufacturer’s protocol. The extracted DNA was quantified using BR dsDNA Assay kit and Qubit fluorometer (Invitrogen, Carlsbad, CA, UK) and PCR-amplified with the 16S rRNA gene V4 region-specific primers following the same procedures as specified earlier by [13].

The purified amplicons were sequenced using the Illumina MiSeq platform (Illumina, San Diego, CA, USA) using 500-cycle v2 chemistry (2×250 bp paired-end reads) at the Centre forEnvironmental Biotechnology, Bangor, UK.

Changed to applying:

The purified amplicons were sequenced using the Illumina MiSeq platform (Illumina, San Diego, CA, USA) applying 500-cycle v2 chemistry (2×250 bp paired-end reads) at the Centre forEnvironmental Biotechnology, Bangor, UK.

L.134-139 Samples of streamer were used for cultivation of eukaryotes. The modified medium 88 (DSMZ) (pH 1.7) containing (g L-1): 1.3 (NH4)2SO4, 0.28 KH2PO4, 0.25 MgSO4 ∙ 7 H2O, 0.07 CaCl2 ∙2 H2O, and solidified with a Gelrite (Sigma), 30 (g L-1), and supplemented with trace element solution SL10 (DSMZ, https://www.dsmz.de/microorganisms/medium/pdf/DSMZ_Medium320.pdf) added at a 1:1000 (v/v) ratio and Kao-Michayluk vitamin solution (Sigma-Aldrich, UK) at a ratio of 1:100 (v/v). To generate axenic cultures, plates were supplemented with ampicillin (100 mg L-1) and carbendazim (20 mg L-1). All plates were grown at 20 °C and subjected to 12 h:12 h light/dark cycles.

The comment: it is really hard to follow.

The text was modified now according to this comment and to the comment of another reviewer as below:

Samples of streamer (5 g) were used for cultivation of eukaryotes. Streamers were washed in 20 ml of the modified medium 88 (DSMZ) (pH 1.7) containing (g L-1): 1.3 (NH4)2SO4, 0.28 KH2PO4, 0.25 MgSO4 ∙ 7 H2O, 0.07 CaCl2 ∙2 H2O. The medium was also supplemented with trace element solution SL10 (DSMZ, https://www.dsmz.de/microorganisms/medium/pdf/DSMZ_Medium320.pdf) added at a 1:1000 (v/v) ratio and Kao-Michayluk vitamin solution (Sigma-Aldrich, UK) at a ratio of 1:100 (v/v). Washed streamers and suspended materials after the wash-off were then sub-divided and plated using sterile loops on to medium 88 (DSMZ) solidified with a Gelrite (Sigma), 30 (g L-1). To generate axenic cultures, plates were supplemented with ampicillin (100 mg L-1) to inhibit the  bacterial growth. The plates were found to have high levels of fungal contamination, therefore subsequent samples were grown on plates also supplemented with carbendazim (40 µg mL-1). All plates were grown at 20 °C and subjected to 12 h:12 h light/dark cycles.

L.155-156 18S rRNA genes were amplified by PCR using MyTaq DNA Polymerase (Meridian Biosciences) using universal eukaryotic primers obtained from [22] EukF Forward Primer (5’-AAC CTG GTT GAT CCT GCC AGT-3’) and EukR Reverse Primer (5’-TGA TCC TCC TGC AGG TTC ACC TAC-3’).

One time “using” was changed:

18S rRNA genes were amplified by PCR with MyTaq DNA Polymerase (Meridian Biosciences) using universal eukaryotic primers obtained from [22] EukF Forward Primer (5’-AAC CTG GTT GAT CCT GCC AGT-3’) and EukR Reverse Primer (5’-TGA TCC TCC TGC AGG TTC ACC TAC-3’).

2.9. Statistical Analysis – should contain the brief description of the analyses performed.

The comment: which, should be specified

Added: All statistical analyses have been carried out using R programming environment [31]. Principal Component Analysis (PCA) has been performed using prcomp function and in-house scripts for graphical design. Multiple regression model was calculated using lm function, with all metal’s concentrations considered as putative predictor variables. From the initial model, the best possible model was chosen using the function stepAIC from the packages MASS [33], selecting the best variables explaining our data for the model. Quantile-Quantile plot was calculated to assess the accuracy of the model and to estimate coefficient and added variable plots were done to analyse specifically the influence of each selected predictor variable (metal) in the model.

L.195 The biogenic streamers at the site are shown in Fig. S1.

The comment: the reference to this Figure should be placed in the subsection 2.1 but not here.

Changed as suggested.

L.199 While the pH values are given for each locality studied, the total carbon and nitrogen values are presented only in average. Why? Are the values in each locality not important for further analysis?

The subsections 3.1 and 3.2 are now combined. The values of the total carbon and nitrogen are important characteristics for acidophiles but were found in relatively similar values in comparison to PM4 site and were measured as an average for the PM5 site.

L.217 It is hard to follow the information in this paragraph, and the percentages unnecessarily given up to hundredths create an additional complication.

Changed: Percentages values are shown up to tenths after the decimal dot. We hope that it is now easier to follow.

L.221 Figure 1. Relative abundances of 16S rRNA gene amplicon reads of the prokaryotic most abundant taxonomic groups (MATGs) across the three subsystems of PM5 site (PM5.1, PM5.2 and PM5.3) and PM4 site, Parys Mt. Circles represent the average relative abundance (%) found among the three replicates of each site. MATGs have been selecting according to a >2% cut off, except for Ca. Micrarchaeota.

The comment: what does it mean - localities?

Fig.1 is Fig.3 now.

Modified, according to the comment:

Relative abundances of 16S rRNA gene amplicon reads of the prokaryotic most abundant taxonomic groups (MATGs) across the three localities of PM5 site (PM5.1, PM5.2 and PM5.3) and PM4 site, Parys Mt. Circles represent the average relative abundance (%) n=3. MATGs have been selecting according to a >2% cut off, except for Ca. Micrarchaeota.

L.122 Done

L.233 Figure 3. Principal Components Analysis (PCA) for the relative abundance of the most abundant taxonomic groups (MATGs) profile shown on each sample site. Ellipses have been calculated over the two different locations (PM4 in red and PM5 in blue), based on mean (dots at the ellipse center) and variance among the samples belonging to each of them. Total relative abundance of each MATG among all samples is represented by the grey rectangles. Expansion scale represents the proportion at which variables (MATGs) axes are drawn regarding the site axes.

The comment: the most abundant taxonomic groups (MATGs), what does it mean?

Fig.3 is Fig.4 now.

Modified to: Principal Components Analysis (PCA) for the relative read numbers of the most-abundant taxonomic groups (MATGs). Any taxon is considered a MATG if it reaches a relative abundance over >=2% of, otherwise its relative read counts are added to the immediate upper lineage level. Once all genera were sorted, the upper level of the lineage was analysed the same way, but without considering the corresponding relative abundance already covered in MATGs. Ellipses have been calculated for the two different locations (PM4 in red and PM5 in blue), based on mean (dots at the ellipse centre) and variance among the samples. Total relative abundance of each MATG among all samples is shown by the grey rectangles. Expansion scale represents the proportion at which variables (MATGs) axes are drawn regarding the site axes.

L.259 Figure 4. Maximum likelihood phylogenetic tree based on 16S rRNA gene sequences of bacteria from sites PM4 and PM5 (PM5.1, PM5.2 and PM5.3), Parys Mt. Relative abundance (%) of reads assigned to taxonomic groups found on each site are shown on the rectangles attached to respective group. Taxonomic groups represented in the tree correspond to the sum of all ASVs assigned to that group. Number of ASVs assigned to each represented group is shown by the size of the grey circle next to each label. Bootstrap score is shown by open (<80) or filled (>80) circles for each node. Scale bar, 0.1 substitutions per position.

The comment “unclear” to: Taxonomic groups represented in the tree correspond to the sum of all ASVs assigned to that group.

Fig.4 is Fig. 6 now

Modified to: Maximum likelihood phylogenetic tree based on 16S rRNA gene sequences of bacteria from sites PM4 and PM5 (PM5.1, PM5.2 and PM5.3), Parys Mt. Relative abundance (%) of reads assigned to taxonomic groups found on each site are shown on the rectangles attached to respective group. Taxonomic groups correspond to the sum of all ASVs by lineage and make groups, taking one ASV sequences as representative and highlighting the number of ASVs on the groups by the size of the circle next to it on the tree. Number of ASVs assigned to each represented group is shown by the size of the grey circle next to each label. Bootstrap score is shown by open (<80) or filled (>80) circles for each node. Scale bar, 0.1 substitutions per position.

L.264

Done

L.280 Taxonomic affiliation.

Please see the feedback to this comment before.

L.281 Changed to: Figure 7. Relative abundance of 16S rRNA gene amplicon reads from principal prokaryotic taxa associated with the streamer growth in PM5.

L.304-306 Please see the feedback to this comment before.

L.312 “contribute” changed to “influence”.

L.316 Changed to: Possibly, the organic carbon exuded into the water from the extensively growing primary producers (e.g., Chlamydomonas acidophila, Euglena mutabilis and Bryopsida sp.) also favours bacterial growth, or their secondary metabolites could be toxic for archaea.

L.320-331 Done

L.338-340 Figure 6. Principal Components Analysis (PCA) representing the abundance profile of the different metals found in the two analysed sites (PM4, red circles; PM5 blue circles), including, additionally, samples from the core surface layers (yellow circles) collected from PM4 sediments [13]. Ellipses have been calculated over the core samples (yellow) and PM5 replicates (in blue), based on mean (dots at the ellipse center) and variance among the samples belonging to each of them. Contribution (%) of each metal to samples separation over PC1 (X axis) is represented by the arrows color (color key at bottom right).

Fig.6 is Fig.1 now.

Changed to: Principal Components Analysis (PCA) representing the abundance profile of different metals present in the two analysed Parys Mt sites (PM4, red circles; PM5 blue circles), including, additionally, samples from the core surface layers (yellow circles) collected from PM4 sediments [13]. Ellipses are drawn for PM4 surface core samples (yellow) for PM5 samples (blue) based on the samples’ variance. Corresponding dots are based on samples’ means. Contribution (%) of each metal to samples separation over PC1 (X axis) is shown by the arrows (colour key is shown in the bottom right).

L.344 and 345. Changed

L.358

The comment: Are the Spanish and Mexican sites are similar to the Parys Mt. sites in terms of their environmental conditions?

All these acidic sites differ in terms of the nature of sites, temperature, solar activity and Eh. TMEG are known to be widely distributed across the range of acidic environments (Korzhenkov et al., 2019).

L.359

Added

L.364

Done

L.369 “and their diversity”

The comment: what does it mean?

Changed to: Sequences clustering with this group were also detected in another Parys Mt study [7], emphasising the ubiquity and the diversity of phylotypes of these organisms in Parys Mt ecosystems.

L.372 Some diversity

The comment: what does it mean?

Changed to: variance

L.374  Italic

The candidate names of taxa should, not be italicised, as per Euzeby nomenclature rules.

L.387 added: water and sediment to: PM4 water and sediment

L.419

Added: In relation to this, the absence of archaea within the streamer points at their other niche preferences in Parys Mt, namely sediments, which was demonstrated previously [13,14].